# Visual Management and Gamification: An Innovation for Disseminating Information about Production to Construction Professionals

Regina Maria Cunha Leite [1], Ingrid Winkler [2,*] and Lynn Rosalina Gama Alves [3]

1 Department of Civil Construction, Federal Institute of Bahia, Salvador 40301-015, Brazil; regina.leite@ifba.edu.br
2 Department of Management and Industrial Technology, SENAI CIMATEC University Center, Salvador 41650-010, Brazil
3 Institute of Humanities, UFBA, Salvador 40170-115, Brazil; lynn@ufba.br
* Correspondence: ingrid.winkler@doc.senaicimatec.edu.br

**Abstract:** The construction industry is undergoing a digital transformation, with the goal of developing long-term solutions that promote construction companies' alignment with market demands and that empower them to reduce production losses as much as possible. The purpose of this paper was to evaluate a gamified model for disseminating production information in the construction industry using visual management. This was a qualitative exploratory study that employed the Design Science methodology and the Design Science Research method. The model was designed, developed, and evaluated by 35 people, including 10 off-site users who focused on usability, user experience, and model promotability, 15 engineers, and 10 workers who considered user experience and promotability. Employees and managers thought the model was excellent, while outside users thought it was good. Furthermore, the evaluators made suggestions for improvements aimed at achieving excellence. We conclude that the proposed model improves production information dissemination in construction by considering the target audience's digital inclusion and knowledge diffusion within work teams.

**Keywords:** gamification; visual management; production; construction

## 1. Introduction

Technological advances, such as the use of mobile devices, social networks, and artificial intelligence, have ushered in a social revolution, enabling previously unimaginable actions to improve quality of life, control, and surveillance, among other things. These changes pose a challenge to the civil construction sector in terms of developing long-term solutions that promote construction companies' alignment with market demands and their ability to reduce production losses as much as possible.

Despite proposals to use digital technologies such as Building Information Modeling (BIM), Big Data, artificial intelligence, and virtual and augmented reality to create a hybrid environment involving people and nonhuman elements such as places, objects, and surfaces [1], construction sites do not reflect this reality and frequently present old production problems, including a lack of transparency in information for workers, labor idleness, unsafe working conditions, and a lack of concern for environmental requirements [2–5].

Bringing the visual management knowledge to construction sites, the lack of transparency contributes to construction production systems underperforming [6]. Instead of performing value-added operations, workers frequently waste time searching, wandering, and waiting for tools, materials, and, especially, information [3].

There are few visual mechanisms in construction companies to inspire, instruct, or motivate workers to perform their jobs more effectively, efficiently, and safely. Structures are required to facilitate worker feedback and to make relevant information transparent

to them [6]. Little feedback on the work being done and the performance of the workers involved in the process leads to low employee engagement on the construction site and makes effectively meeting production goals difficult [7].

In recent years, empirical studies on visual management at the construction site focused on the use of the Digital Spreadsheet (Excel) [5], Dashboards with charts [8], BIM Modeling, 4D BIM and 3D mappings [9], the SyncLean prototype [10], and the Digital Room to share information [11], whereas Refs. [3,4,12,13] used A3 paper sheets and post-its to share information. Some studies [3,4,10,12,13] aimed to promote transparency to improve worker understanding, while others [5,8,9,11] focused on the promotion of information transparency for engineers.

According to the studies, understanding the information about the work to be developed, collaboration, and teamwork are critical components to achieving the goals established in the planning. The delivery of information appears to be accelerated by digital technology. However, without the collaboration and integration of teams at various levels, the success and benefits of digital technology use do not occur, jeopardizing Kaizen (continuous improvement), which is the result of dedication and learning from previous experiences.

In this context, gamification, defined as the use of game-based mechanics, dynamics, and thinking in a real-life context to develop skills, motivate actions, promote learning, and solve problems [14–19], is an attractive prospect for the construction industry to improve communication between workers and engineers. The strategy proposes motivating employees to perform repetitive tasks, making the workplace more collaborative, and encouraging professionals to adopt player behavior, i.e., proactive or reactive, behavior, to achieve set goals.

Few empirical studies on the use of gamification on construction sites have been conducted. An exploratory literature review identified only four studies conducted between 2014 and 2020: Neto et al. [7] presented a plan for implementing a gamified system in a construction site to make weekly planning information transparent to the company, construction team, and workers; Leite et al. [20] presented a three-month implementation of a gamified system in a construction site to make weekly planning information transparent to the company, construction team, and professionals.

For four months, Khanzadi et al. [21] developed and tested a gamified system for applying lean to production planning and control in offshore construction, and Selin et al. [22] presented a gamified method for planning the safety of information modeling (BIM)-based environments. These are intended for engineers who work with the systems. Khanzadi et al. [21] proposed financial incentives for engineers that are paid directly from the organization's payroll, and they note that the proposal was not well received by the organization. Khanzadi et al. [21], as well as Selin et al. [22], proposed using gamified systems to motivate participants and increase productivity.

Neto et al. [7] and Leite et al. [20] targeted construction professionals. They emphasized the importance of employee engagement, transparency of relevant information, and feedback for those involved, but they did not address ethical issues and instead relied solely on points, badges, and leaderboards (PBL) to promote engagement. Schlemmer [23] defined PBL as an empiricist educational technique that reduces gamification to a fad, something superficial, and with low innovation power. It is the 'shell of a game experience', according to Chou [24], accounting for only 7% of the total tactics mapped by the Yu-Kai-Chou Octalysis Framework.

According to Alves and Souza [25], gamification in the PBL perspective impoverishes and limits the possibilities of creating rich, contextualized narratives that reflect content that mobilizes and engages subjects, as in the case of production gamification, where workers enjoy soccer. Thus, with its narratives and missions, gamification on construction sites has yet to be explored and may be a solution to improve communication and motivate construction workers to meet their production goals.

As a result, the goal of this paper is to assess a gamified model that employs visual management as a mechanism for disseminating production information to construction professionals.

It is hoped that the gamified environment will make work more interesting for both workers and engineers. The former can be motivated by a sense of belonging, of being part of a group and working together to achieve goals, of being recognized for their work, and of participating in solutions. In the case of engineers, satisfaction from meeting demands efficiently and with quality is assumed to be a positive incentive for interacting with such environments.

This paper is organized as follows: Section 2 describes the materials and methods used, Section 3 presents and analyzes the results, and Section 4 presents our conclusions and suggestions for further investigation.

## 2. Materials and Methods

The Design Science methodological approach is recommended in research that aims to create, develop, and explore new solutions [26]. The Design Science Research method was used for this study, which is concerned with the creation and evaluation of artifacts with the goal of solving real-world problems [27–29].

This was an exploratory, empirical, and qualitative study aimed at learning about gamification and visual management strategies as mechanisms for improving communication and dissemination of production information to construction professionals.

The proposed model's functionalities and usability were evaluated with the help of 35 people, including 10 workers, 15 engineers, and 10 external construction site users.

The researcher mentored the trainee in conducting the training and interviews. During the data collection phase of this same project, he was a member of the research team at the same company.

The model was presented to the workers, demonstrating the tool's functionality, the meaning of the icons, team feedback, and individual feedback. The training lasted approximately 30 min. Following that, everyone was given a WhatsApp link to use the system. This occurred following the weekly construction site meeting, during which the trainee was available to answer any questions about the system. Beginning with the next weekly meeting, the trainee interviewed two workers per day, gathering their feedback on the model using a form that combined the UEQ-S and NPS tools (Figure A1). The interviewer read the form's questions and alternative answers and marked the one selected by the interviewee. Only ten workers were performing regular services at this time due to the pandemic and the mandatory reduction in the number of employees on site to maintain social distancing.

A total of 70 engineers and 50 external users (computer science, building, engineering, and management students, business administrators, and engineers) were invited to participate in the engineers and external users' evaluation. Only 15 engineers and ten outside users agreed. A link to the system was made available for this public in the system's public characteristics as well as in the private links of a fictitious functionary, for model analysis and information flow. Because the system is simple and clear, the training was limited to a brief explanation of the tool's functionality and the meaning of the icons.

Engineers were also instructed to complete the questionnaire using the UEQ-S and NPS tools following system testing (Figure A2).

To assess the external users of the construction site, a questionnaire containing the SUS, UEQ, and NPS tools (Figures A3 and A4) was distributed to people who are familiar with the construction process but do not work in the area. The sample consisted of one student from the technical building course (IFBA), three undergraduate civil engineering students (UFBA), two business administrators, and four engineers from the project area of construction companies.

All participants were given one week to use the system and complete the questionnaires (see Supplementary Materials).

The UEQ-S tool is a simplified full UEQ model that asks for the evaluation of eight constructs to determine the product's quality according to the user. The product's quality can be measured using a Likert scale ranging from −3 to +3. Products with a quality rating of less than −0.8 are not recommended [30–32].

We used the questionnaires and analysis tools available on the User Experience Questionnaire website to generate data and perform the evaluation calculations [30]. The UEQ has 26 items divided into six scales: attractiveness, transparency, efficiency, control, stimulation, and innovation [31]. Each item represents a diametrically opposed pair: complicated/easy, conservative/innovative, or fast/slow. The same is true for the UEQ-S (Figure 1), which consists of 8 items divided into two scales: Hedonic Quality and Pragmatic Quality.

| Negative | 1 | 2 | 3 | 4 | 5 | 6 | 7 | Positive |
|---|---|---|---|---|---|---|---|---|
| obstructive | | | | | | | | supportive |
| complicated | | | | | | | | easy |
| inefficient | | | | | | | | efficient |
| confusing | | | | | | | | clear |
| boring | | | | | | | | exciting |
| not interesting | | | | | | | | interesting |
| conventional | | | | | | | | inventive |
| usual | | | | | | | | leading edge |

https://www.ueq-online.org/

**Figure 1.** User Experience Questionnaire (UEQ-S). https://www.ueq-online.org/ (accessed on 13 January 2020).

The Promotion Score NPS tool asks the user if they would recommend the product to a friend or colleague. This is an indirect method of determining how much the user trusts the system. Despite being composed of a single question, its main goal is to determine whether the user would recommend the product [32].

This index was used to determine whether the subject trusted the model enough to recommend it to a friend or coworker. In this case, instead of using a scale from 0 to 10, responses were consolidated into three groups: (0–6)—detractors, (7–8)—passives, and (9–10)—promoters. The answer was NO for the detractors, MAYBE for the supporters, and YES for the promoters. Finally, the NPS was calculated, which is equal to the percentage of promoters minus the percentage of detractors and is classified as follows: Excellence Zone (76–100), Quality Zone (51–75), Improvement Zone (0–50).

Usability/Ease of learning the interface: Usability is linked to the ease of learning and using the interface, as well as user satisfaction as a result of this use. The System Usability Scale (SUS) is made up of ten statements on a Likert scale, with odd-numbered questions having a positive connotation and even-numbered questions having a negative connotation. To account for the outcome, the answers are assigned values ranging from 0 to 4; for the odd ones, 1 is subtracted from the informed value, and for the even ones, the result is reduced by 5, adding all the results and multiplying by 2.5 to obtain a score [33].

Because the research is qualitative, these methods complement one another. The evaluator uses a variety of methods to determine what is good and what can be improved in the system.

## 3. Results and Discussion

In the following subsection, we present and analyze our results.

### 3.1. The Gamification Model in Production

For implementing the model, we used cloud architecture that provides a set of flexible interaction consoles, data monitoring and storage, and data privacy management. The gamified web system was developed in Python using the Django framework. The main objective was to make the weekly goals more transparent to the entire site team and to

display the teams' performance and weekly service progress on a worksite monitor. The context of a soccer championship was used as an engagement strategy (narrative), because approximately 60% of professionals enjoy and participate in this sport.

Site supervisors are usually responsible for monitoring the services as well as individual and team performance. They can enter the information manually during supervision using the existing interface or integrate their management systems with the gamified system so that the data are entered automatically. In addition, the worker can receive feedback on their performance based on the data collected during service monitoring via a private link sent via WhatsApp.

For the system to function properly, rules, challenges, and missions should be developed in collaboration with the site engineers during the implementation process, so that the gamified system can display the performance of teams and workers. To motivate employees, the gamification strategy proposed here employs points, challenges, missions, avatars, badges, and positive messages in the form of a soccer championship. This work is based on the Empowerment model, which promotes autonomy, collaboration, and cooperation in groups through narratives, missions, challenges, and discoveries [34].

'Motivational factors for employees, such as identifying top performing employees and displaying their performance on display boards and making them visible among other employees that encourage others to perform well', according to Subhav, were recommended [35] (p. 1161). However, the display in this case will be for teams, and workers will only have access to their own performance for ethical reasons; additionally, tracking the worker on the construction site using sensors or cameras is not possible, so individual feedback will not be visible in real time.

The Brazilian Soccer Championship inspired the development of gamification. In comparison, the clubs in the Brazilian Championship correspond to labor teams; the season, which runs from April to December each year, has timeframes that can be configured at the discretion of the site management. The rounds in both correspond to one week, which corresponds to the project's short-term programming.

Because there are team changes and new services, the duration of the championship is configurable depending on the stage of the project. There is no relegation in this championship, and even if the team does not score a goal, it can still score points.

Because it is important not to distract the workers who perform manual labor on the construction site, the gamification results are modified based on the data entered by the supervisor of the services performed by the teams throughout the week. Similarly to the Brazilian Championship, the games are played on Sundays or at night, once a week, when the public can watch, and the championship outcome changes with each game.

The services completed, in progress, or just begun will be visualized on the construction site monitor or on the worker's smartphone, using the colors green, yellow, and red to represent good, reasonable, and bad, which is the symbology already known to all.

Figures 2 and 3 depict illustrative system screens that show the weekly goals and performance of each team.

The following progress levels were assigned in the example shown (Figure 2): green—indicating that the team has completed more than 90% of the service; yellow—indicating that the team has completed more than 50% of the job; and red—indicating that the team has completed less than 50% of the job.

Figure 3 depicts the teams' performance screens, which use icons of the same colors (green, yellow, and red) to indicate not only progress, but also the following scores: 20 points for a ball in the net, 12 points for a ball at the post, and 4 points for a ball out.

Weekly, the total score is displayed in points. The number of points earned determines who wins the trophy for the week.

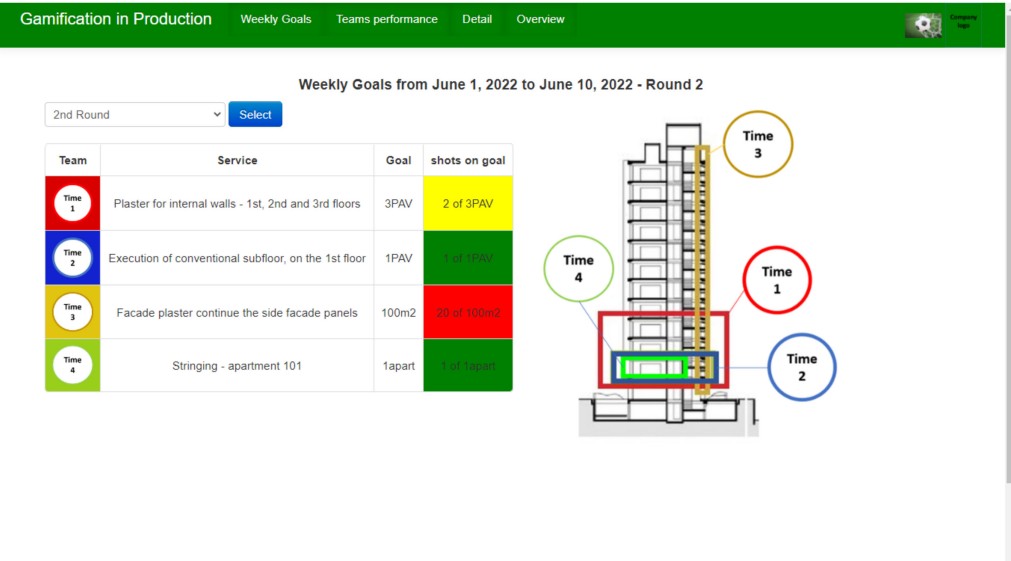

**Figure 2.** Weekly goals screen.

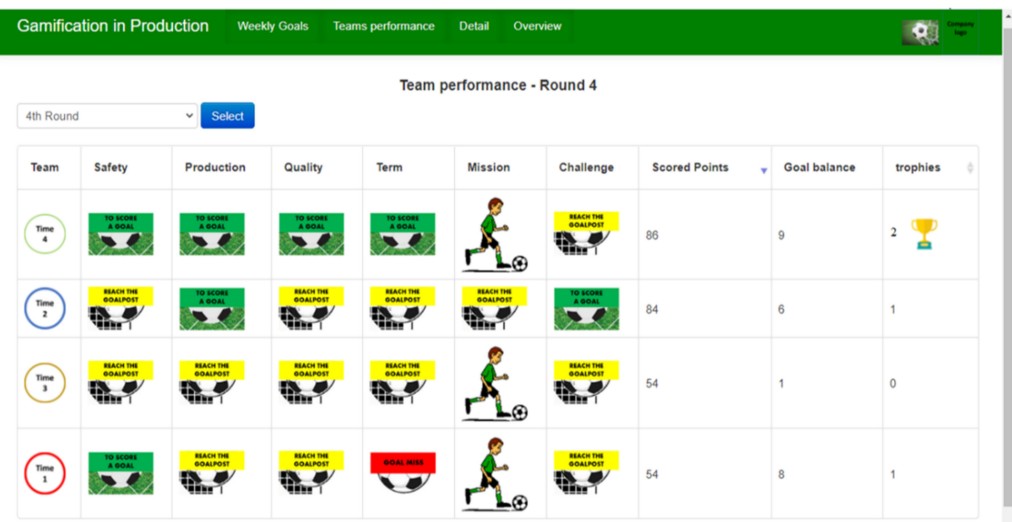

**Figure 3.** Team performance screen.

There is also the goal difference, which accumulates how many goals (ball in the net) the team has scored each week throughout the championship. The goal difference determines who wins the championship. Each team is represented by an avatar, in this case, the clubs of the Brazilian Championship, but nothing prevents them from using other symbols to identify themselves.

Goals with the names Safety, Quality, Deadline, and Production were created here, which are related to the service provided by the team and can assume the statuses represented by the icons.

When planning the gamification process, the goals, challenges, and missions will be developed in collaboration with the engineers. The challenges are timed tasks that must be completed throughout the day by a specific deadline. An example of an initial challenge that can be implemented is to select a song to play every time the team wins the trophy of the week, that is, to achieve the most goals. These celebrations can take place in the cafeteria or during the site's weekly meeting. Another example is a team selecting an image to represent itself (avatar). Always keep in mind that the challenge is something that can be completed quickly and with a limited amount of time.

The missions, on the other hand, can be completed at any time during the week. A mission could be something like this: some companies provide school to their employees outside of office hours, or computer courses to those who already have a basic level. As a result, enrolling at least one team member in these courses would be a mission.

In addition, the system includes Detail and Overview functions. Short training videos can be posted on the construction site using the Detail function. Consider the explanation of the shortest path to minimize transportation losses in construction processes [36], as well as the use of simulations in BIM 4D to present the most appropriate metal form assembly planning method, resulting in cycle time reduction [37] as examples of these short trainings. The overview feature displays important images for the stage of the work, such as floor plans of the floor being executed and team location plans.

To demonstrate individual feedback, we created a fictitious worker named Philip who received his private link via WhatsApp to access information about his individual performance, as depicted in Figure 4.

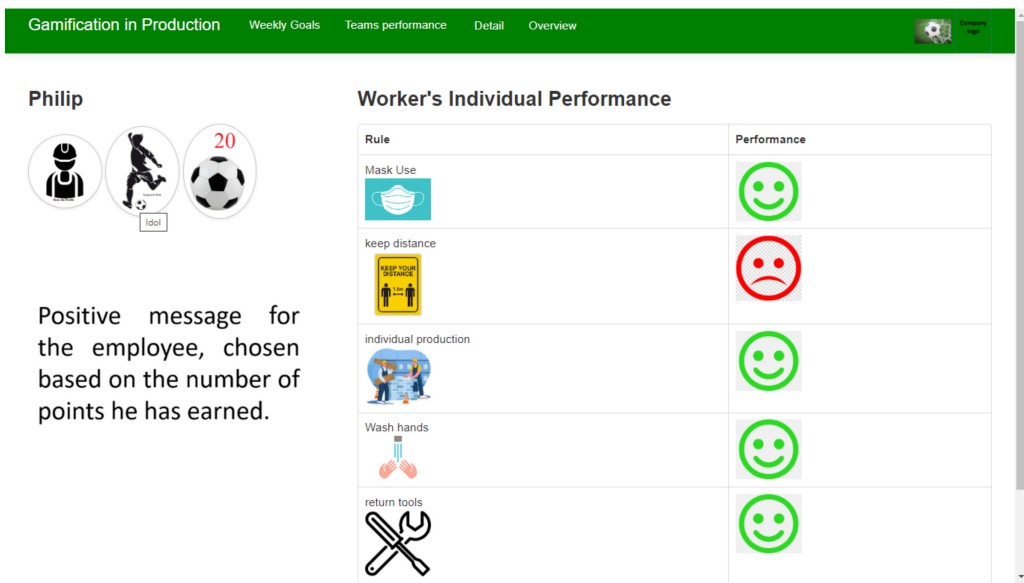

**Figure 4.** Worker performance through the private link.

To improve communication, the worker can see individual attitudes (return tools, wear mask, wash hands, etc.) represented by pictures on this screen. His performance was represented by green, yellow, and red emoji. Figure 4 depicts a profile picture, a picture of the soccer idol, an emblem with the number of points earned during the week, a ball with the number 20, and a positive message (quote from a soccer star).

This model's information is fictitious and was added for testing and evaluation purposes.

### 3.2. Evaluation of the Gamification Model in Production

The following requirements must be met before this gamified system can be implemented on the construction site:

1.  The company must have a well-structured weekly planning process in place, as well as daily production information for its services.
2.  Install a 32-inch monitor in a strategic location on the job site to display system information.
3.  Have a computer connected to a monitor from which the web system will be accessed.
4.  Having enough capacity for Wi-Fi that can be accessed from anywhere on site.

In short, the Production Gamification system presents the services and goals to be developed and achieved during the week using a soccer championship metaphor. On the first screen, the system displays the services, quantities, and teams that will develop them. Throughout the week, the 'Shots on goal' field changes color, becoming red for services that

are less than 50% complete, yellow for those that are more than 50% complete, and green for those that are more than 90% complete. The second screen displays the teams' performance in four goals, such as safety and production, as well as a challenge and mission that will be presented at the start of the week.

The team with the most points wins the weekly trophy.

The third screen, accessed via a private link, displays the performance of each individual worker.

The second and third screens provide feedback on how the services were developed to the teams and workers.

Figure 5 depicts how visual management and gamification should be used on the construction site to improve information flow and suggests scenarios in which functions should be used.

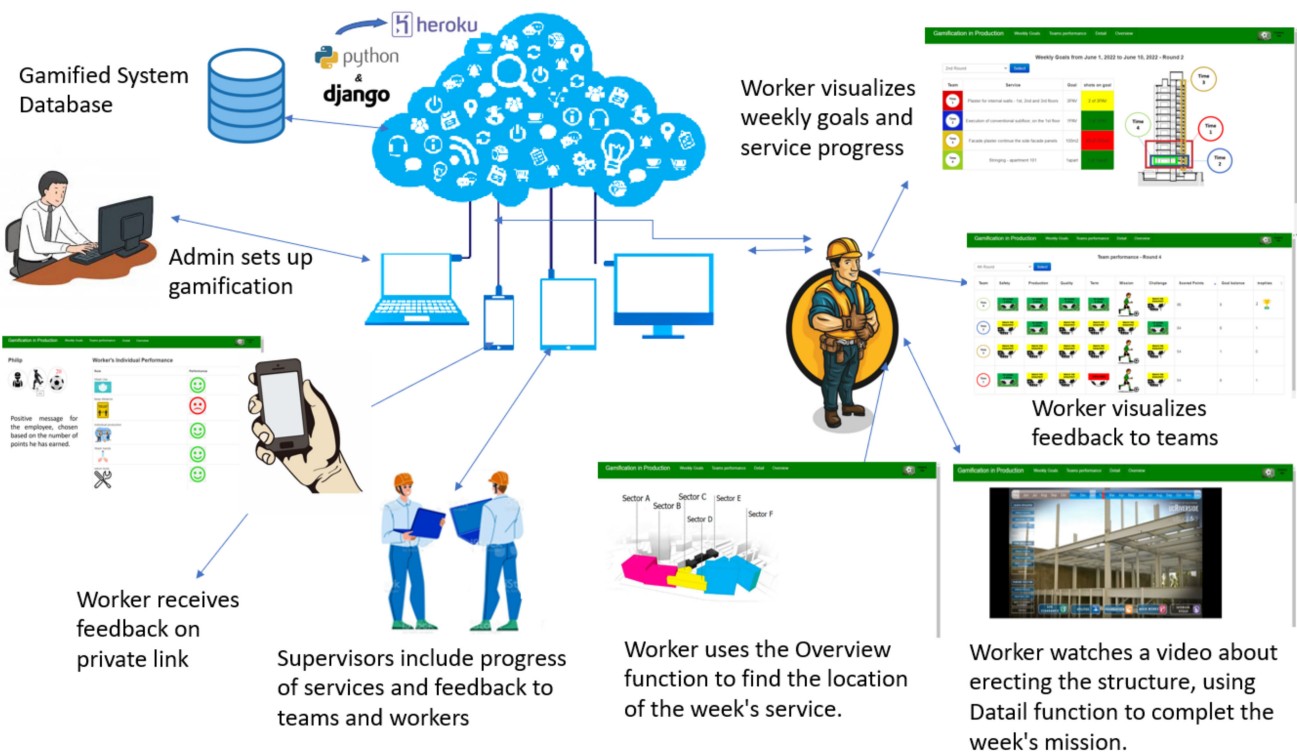

**Figure 5.** Information flow in construction site.

Each subject of this study evaluated the model in production using data generated by the instruments outlined in the methodology. The model representing not only the gamified system, but also the entire environment created by its use, was considered for evaluation. Engineers, workers, and users evaluated the model's functionality and usability.

### 3.2.1. Analysis of the Gamified Model from the Workers' Answers

The model was presented to the worker, demonstrating the tool's functionalities as well as how team and individual feedback work. Following the presentation, a form was used to collect responses on the model. This form is broken down into the following sections: worker profile, consent for research participation, UEQ-S tool, NPS, and questions about the worker's preferences. Figures 6 and 7 show the outcomes.

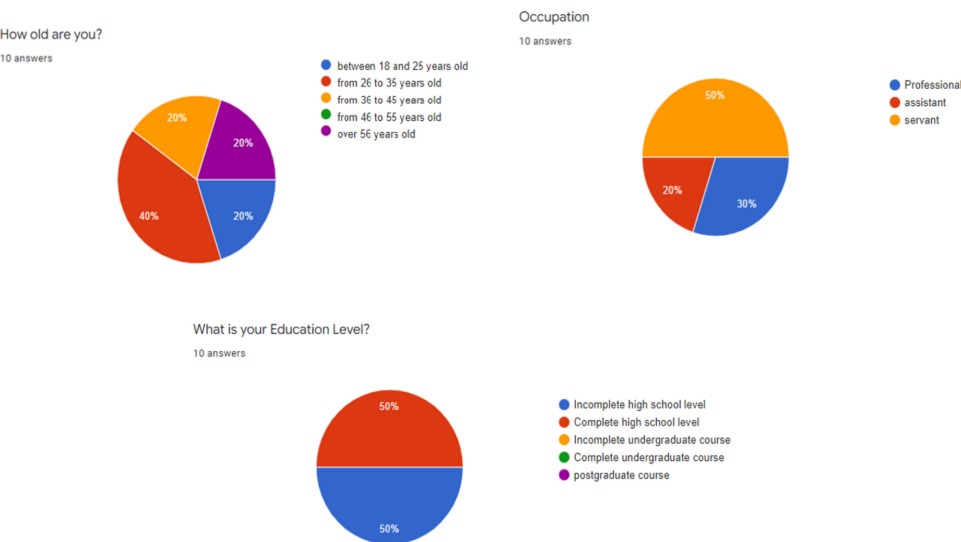

**Figure 6.** Profile of the sample of workers.

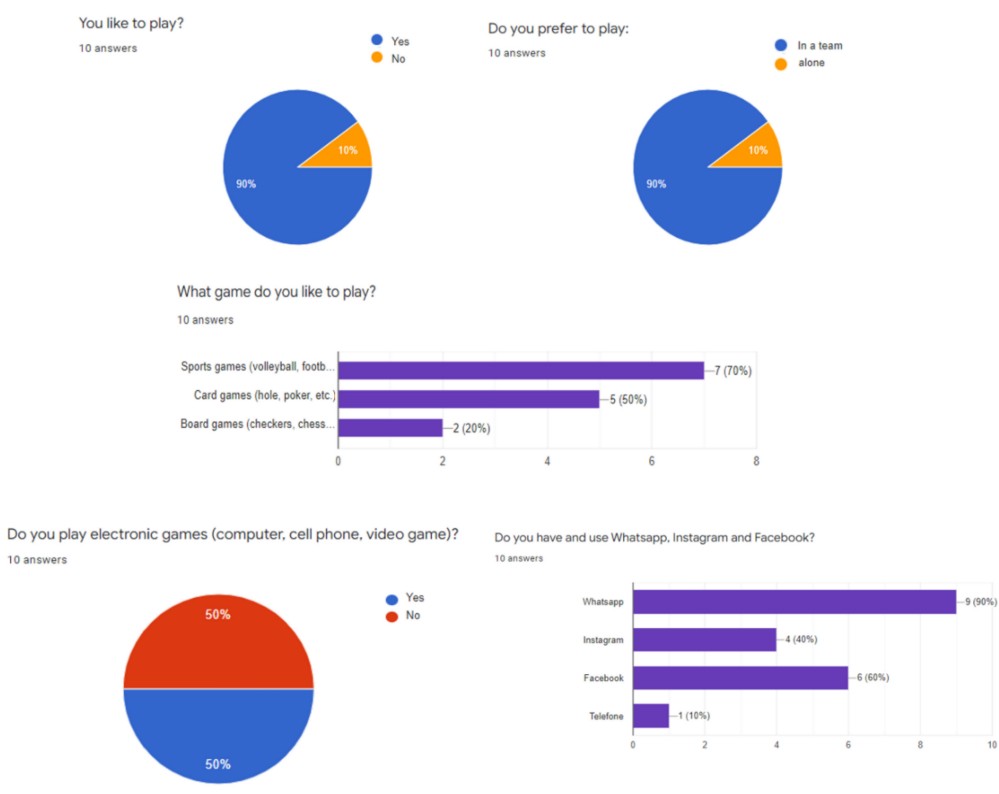

**Figure 7.** Workers x Games x Technology.

As a result, as illustrated in Figure 6, 60% of the workers are young (less than 36 years old), 50% are servants, and 50% have an incomplete middle school education.

An amount of 90 percent of them enjoy playing, with 90 percent preferring team games, and 50 percent preferring electronic games. They use social media, and 90 percent of them have WhatsApp.

Table 1 summarizes the workers' answers to the UEQ-S.

**Table 1.** Workers' answers to the UEQ-S.

| Item | Mean | Variance | Std. Dev. | No. | Negative | Positive | Scale |
|------|------|----------|-----------|-----|----------|----------|-------|
| 1 | 2.5 | 0.5 | 0.7 | 10 | obstructive | supportive | Pragmatic Quality |
| 2 | 3 | 0 | 0 | 10 | complicated | easy | Pragmatic Quality |
| 3 | 3 | 0 | 0 | 10 | inefficient | efficient | Pragmatic Quality |
| 4 | 2.7 | 0.9 | 0.9 | 10 | confusing | clear | Pragmatic Quality |
| 5 | 2 | 1.6 | 1.2 | 10 | boring | exciting | Hedonic Quality |
| 6 | 2.9 | 0.1 | 0.3 | 10 | not interesting | interesting | Hedonic Quality |
| 7 | 2.3 | 3.6 | 1.9 | 10 | conventional | inventive | Hedonic Quality |
| 8 | 2.2 | 3.5 | 1.9 | 10 | usual | leading edge | Hedonic Quality |

All scales were made up of a series of seven-point items ranging from −3 (extremely bad) to +3 (extremely good). Values between −0.8 and 0.8 indicate a neutral evaluation of the corresponding scale, values greater than 0.8 indicate a positive evaluation, and values less than 0.8 indicate a negative evaluation [30].

Figure 8 shows that the model has positive characteristics in the perception of workers, such as being interesting to receive feedback through gamification, reaching values above 2.0 (two), indicating that they approved of the model's quality and believe it will contribute to optimizing their daily work. For workers, pragmatic quality (efficiency, ease of use, and dependability) is more important than hedonic quality (originality, stimulation), both of which received high ratings.

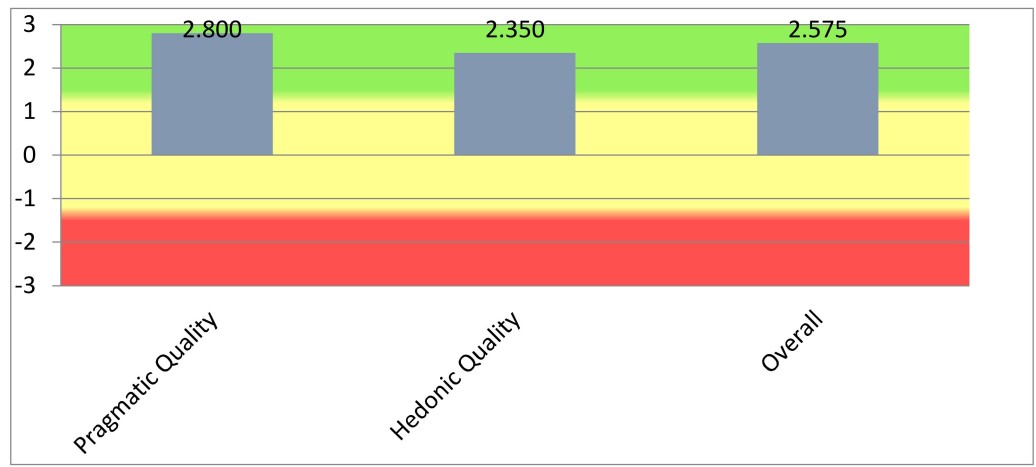

**Figure 8.** Quality of the model in the perception of the workers.

A benchmarking database is included with the UEQ-S tool. The averages of the measured scale are defined in relation to the existing values of a benchmark dataset. This dataset includes data from 14,056 people from 280 studies on various products (business software, web pages, web stores, social networks). When the evaluated product's results are compared to the benchmark data, conclusions about the relative quality of the evaluated product in comparison to other products can be drawn. These findings are shown in Table 2.

**Table 2.** Benchmark results for workers' responses to the UEQ-S.

| Scale | Mean | Comparison to Benchmark | Interpretation |
|-------|------|-------------------------|----------------|
| Pragmatic Quality | 2.80 | Excellent | In the range of the 10% best results |
| Hedonic Quality | 2.35 | Excellent | In the range of the 10% best results |
| Overall | 2.58 | Excellent | In the range of the 10% best results |

The NPS was evaluated using only one question: Would you recommend this model for use on the site where you work? (Figure 9).

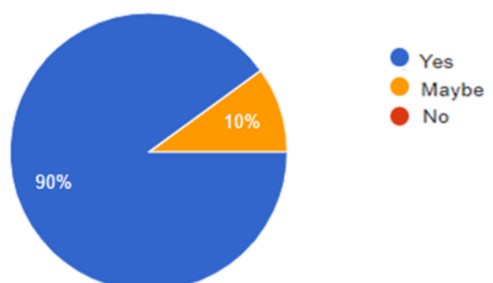

**Figure 9.** Promotion of the model (NPS) to the worker.

Nine of the ten workers said 'yes', with only one saying 'maybe', indicating that the model is in the Zone of Excellence (76–100) for this audience.

### 3.2.2. Analysis of the Gamified Model Based on the Engineers' Answers

Analyzing the profiles of the 15 engineers who responded, 67 percent of the sample is made up of young engineers, between the ages of 26 and 35, and 35.3 percent of engineers are over the age of 56, indicating a diverse range of experience, as shown in Figure 10.

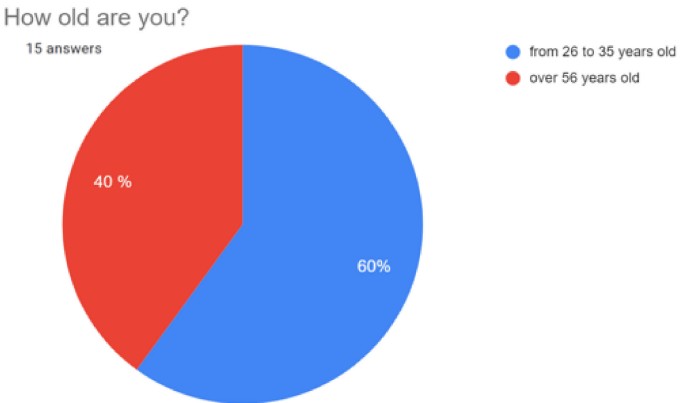

**Figure 10.** Profile of the sample of construction engineers.

After the engineers' evaluation of the model, a summary of the answers is presented in Table 3.

**Table 3.** Summary of the engineers' answers to the UEQ-S.

| Item | Mean | Variance | Std. Dev. | No. | Negative | Positive | Scale |
|------|------|----------|-----------|-----|----------|----------|-------|
| 1 | 1.8 | 2.5 | 1.6 | 15 | obstructive | supportive | Pragmatic Quality |
| 2 | 2.0 | 1.4 | 1.2 | 15 | complicated | easy | Pragmatic Quality |
| 3 | 2.2 | 0.5 | 0.7 | 15 | inefficient | efficient | Pragmatic Quality |
| 4 | 2.1 | 0.9 | 1.0 | 15 | confusing | clear | Pragmatic Quality |
| 5 | 2.2 | 0.5 | 0.7 | 15 | boring | exciting | Hedonic Quality |
| 6 | 2.6 | 0.4 | 0.6 | 15 | not interesting | interesting | Hedonic Quality |
| 7 | 2.6 | 0.5 | 0.7 | 15 | conventional | inventive | Hedonic Quality |
| 8 | 2.3 | 1.1 | 1.0 | 15 | usual | leading edge | Hedonic Quality |

Table 3 demonstrates that the model has positive characteristics, as measured by values greater than 1.8, in the engineers' opinion. Engineers rate pragmatic quality (efficiency, ease of use, dependability) lower than hedonic quality (originality, stimulation), indicating how open they are to gamification on the construction site. According to the evaluations for general quality above 2.0, they believe that the solution will motivate workers to achieve their goals, as shown in Figure 11.

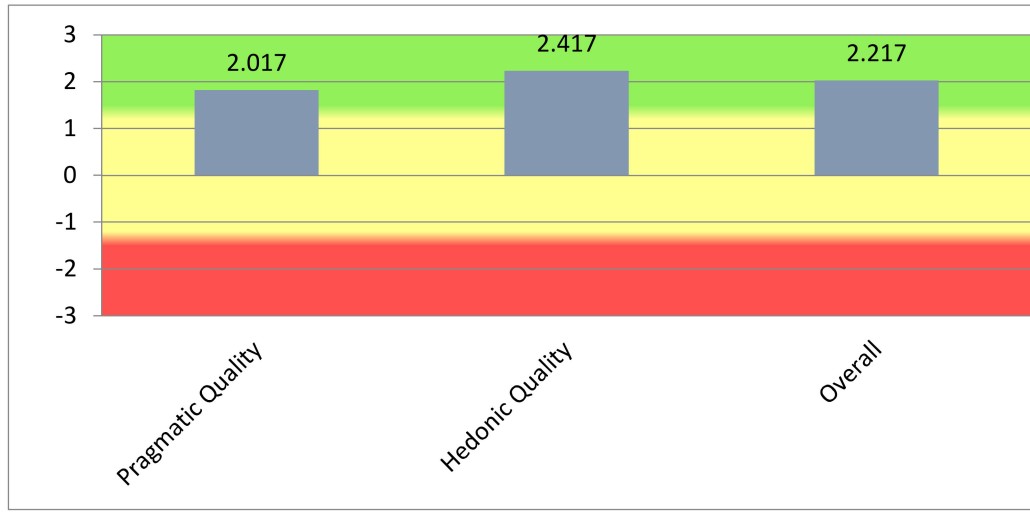

**Figure 11.** Quality of the model as perceived by engineers.

When the evaluated product's results are compared to the reference data, conclusions about the evaluated product's relative quality can be drawn. Table 4 summarizes these findings.

**Table 4.** Benchmark results for engineers' responses to the UEQ-S.

| Scale | Mean | Comparison to Benchmark | Interpretation |
|---|---|---|---|
| Pragmatic Quality | 2.02 | Excellent | In the range of the 10% best results |
| Hedonic Quality | 2.42 | Excellent | In the range of the 10% best results |
| Overall | 2.22 | Excellent | In the range of the 10% best results |

Regarding NPS, all engineers would recommend the gamified model, which demonstrates that the model is in the Zone of Excellence (76–100) for engineers, as shown in Figure 12.

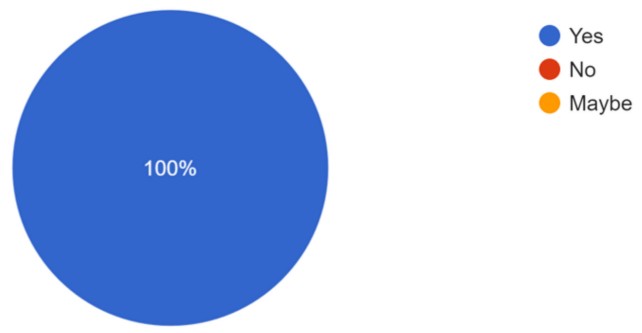

**Figure 12.** Model promotion (NPS) for the engineers.

### 3.2.3. Analysis of the Gamified Model from the Responses of Off-Site Users

Figure 13 depicts responses from people of varying ages and education levels, with a focus on young people.

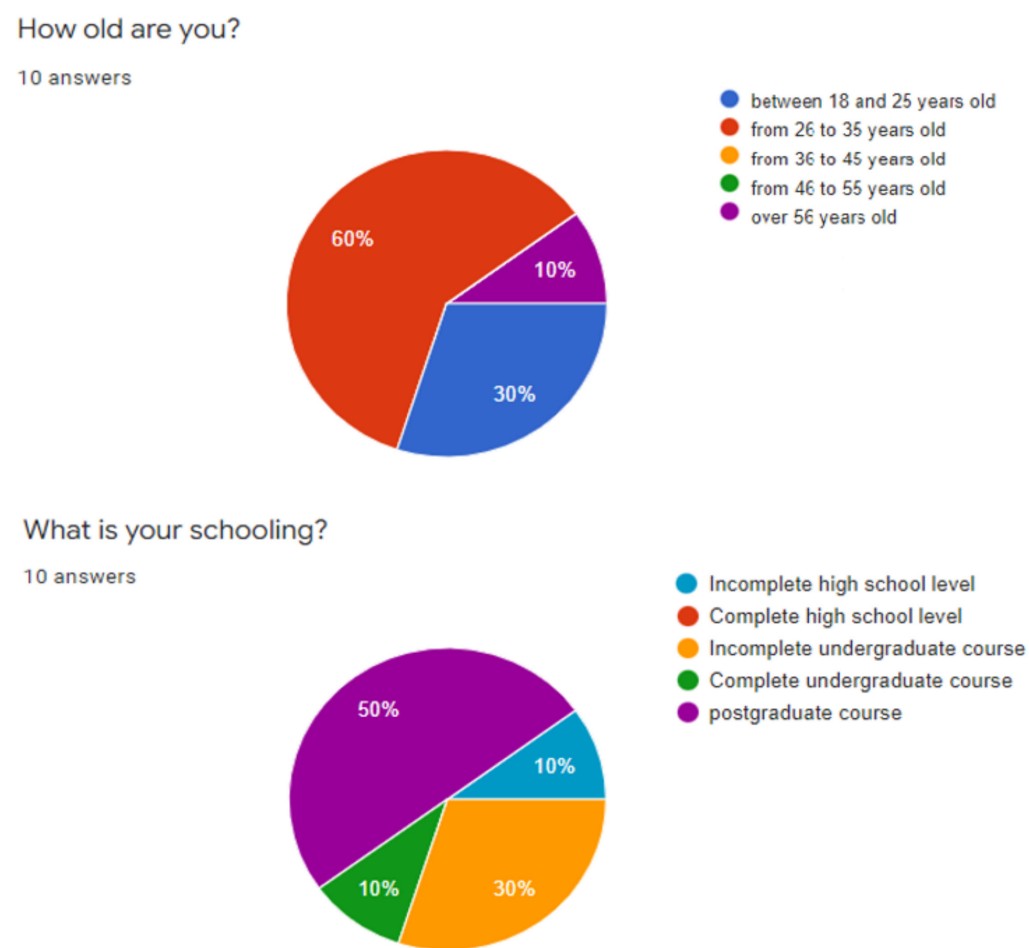

**Figure 13.** Age range and education level of external users.

The first section of the questionnaire refers to the System Usability Scale tool, which assesses the system's acceptability and is composed of 10 statements on which the respondent must indicate their level of agreement on a scale of 0 to 4, as explained in the methodology. As a result of using this tool, the averages were calculated, and a value of 61 was obtained, indicating acceptability as an average (50–70).

Because they were not a specific construction production audience, the external user group had some difficulty understanding the system, indicating that care should be taken in selecting the images representing the weekly goals and that training workers in the system are essential for their understanding.

To evaluate the gamified model based on the information displayed on the worker screen, external users were asked to rate it using the full UEQ Tool, with flag values ranging from 1 (most negative) to 7 (most positive) for each of the attributes. Table 5 contains a summary of the responses.

**Table 5.** Summary of external users' responses to the UEQ.

| Item | Mean | Variance | Std. Dev. | No. | Left | Right | Scale |
|------|------|----------|-----------|-----|------|-------|-------|
| 1 | 2.1 | 1.0 | 1.0 | 10 | annoying | enjoyable | Attractiveness |
| 2 | 1.4 | 4.0 | 2.0 | 10 | not understandable | understandable | Perspicuity |
| 3 | 0.0 | 7.1 | 2.7 | 10 | creative | dull | Novelty |
| 4 | 1.0 | 4.9 | 2.2 | 10 | easy to learn | difficult to learn | Perspicuity |
| 5 | 0.5 | 6.1 | 2.5 | 10 | valuable | inferior | Stimulation |
| 6 | 1.8 | 0.8 | 0.9 | 10 | boring | exciting | Stimulation |
| 7 | 2.4 | 0.5 | 0.7 | 10 | not interesting | interesting | Stimulation |
| 8 | 1.1 | 0.8 | 0.9 | 10 | unpredictable | predictable | Dependability |
| 9 | 0.4 | 7.2 | 2.7 | 10 | fast | slow | Efficiency |
| 10 | 1.3 | 7.6 | 2.8 | 10 | inventive | conventional | Novelty |
| 11 | 2.4 | 0.9 | 1.0 | 10 | obstructive | supportive | Dependability |
| 12 | 1.1 | 7.0 | 2.6 | 10 | good | bad | Attractiveness |
| 13 | 2.2 | 1.5 | 1.2 | 10 | complicated | easy | Perspicuity |
| 14 | 2.4 | 0.5 | 0.7 | 10 | unlikable | pleasing | Attractiveness |
| 15 | 2.1 | 1.4 | 1.2 | 10 | usual | leading edge | Novelty |
| 16 | 1.9 | 1.7 | 1.3 | 10 | unpleasant | pleasant | Attractiveness |
| 17 | 1.1 | 4.3 | 2.1 | 10 | secure | not secure | Dependability |
| 18 | 0.9 | 6.5 | 2.6 | 10 | motivating | demotivating | Stimulation |
| 19 | 1.5 | 4.9 | 2.2 | 10 | meets expectations | does not meet expectations | Dependability |
| 20 | 2.0 | 1.6 | 1.2 | 10 | inefficient | efficient | Efficiency |
| 21 | 0.6 | 4.3 | 2.1 | 10 | clear | confusing | Perspicuity |
| 22 | 2.2 | 1.1 | 1.0 | 10 | impractical | practical | Efficiency |
| 23 | 1.5 | 3.6 | 1.9 | 10 | organized | cluttered | Efficiency |
| 24 | 1.1 | 3.0 | 1.7 | 10 | attractive | unattractive | Attractiveness |
| 25 | 1.3 | 6.0 | 2.5 | 10 | friendly | unfriendly | Attractiveness |
| 26 | 2.8 | 0.2 | 0.4 | 10 | conservative | innovative | Novelty |

Figure 14 shows that the external users' results are all positive and greater than 0.8.

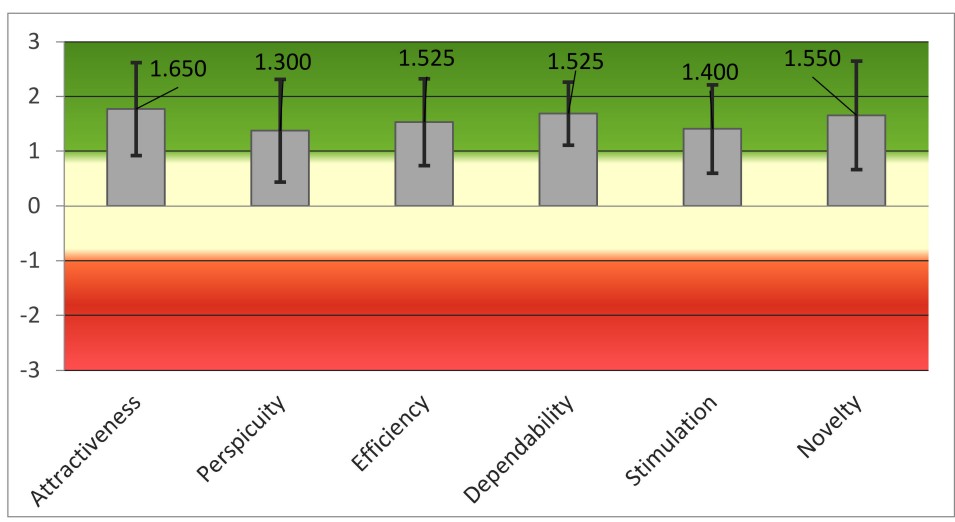

**Figure 14.** Graph of the external users' results to the UEQ.

The UEQ tool also allows us to compare the model's attractiveness and quality. For external users, we observed that the model's Pragmatic and Hedonic Quality are very close positive values, well above 0.8, which the tool's authors consider the neutral range limit [30]. They assigned a higher value to Attractiveness than to Pragmatic and Hedonic Quality (Figure 15), implying that the model is attractive, 'good' in task-related quality aspects, and interesting in non-task-related quality aspects.

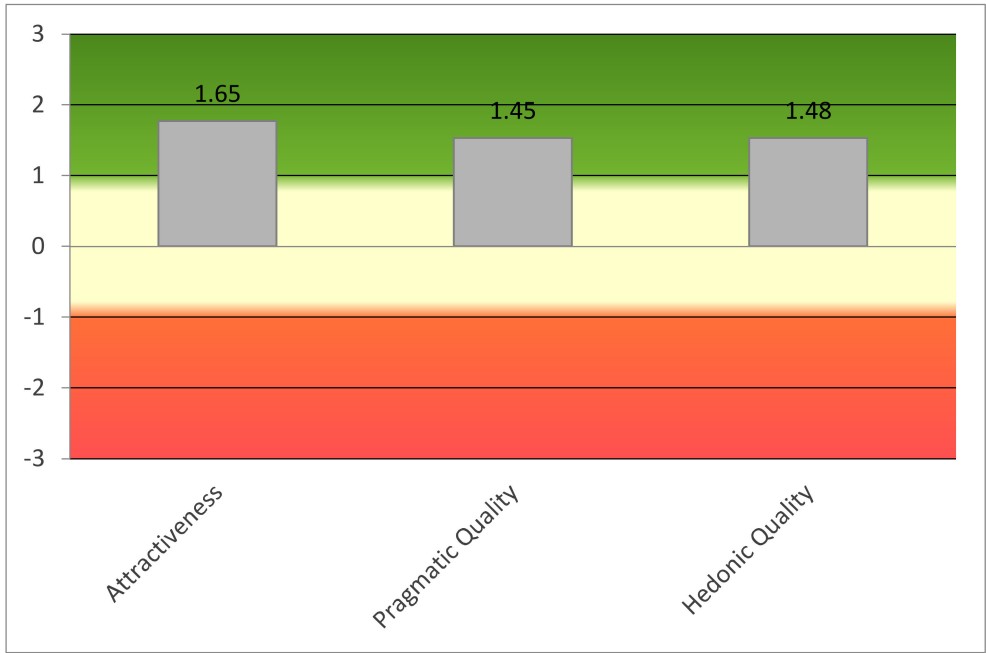

**Figure 15.** Attractiveness, pragmatic quality, and hedonic quality.

As a result, external users rated the model as 'Good' in all measured indexes: attractiveness, transparency, efficiency, control, stimulation, innovation, pragmatic quality, and hedonic. The results of the evaluated product were compared to the benchmark data, as in previous analyses. Table 6 displays these findings.

**Table 6.** Comparison to benchmark.

| Scale | Mean | Comparison to Benchmark | Interpretation |
|:---:|:---:|:---:|:---:|
| Attractiveness | 1.65 | Good | 10% of results better, 75% of results worse |
| Perspicuity | 1.30 | Above Average | 25% of results better, 50% of results worse |
| Efficiency | 1.53 | Good | 10% of results better, 75% of results worse |
| Dependability | 1.53 | Good | 10% of results better, 75% of results worse |
| Stimulation | 1.40 | Good | 10% of results better, 75% of results worse |
| Novelty | 1.55 | Good | 10% of results better, 75% of results worse |

In terms of the NPS, we also attempted to determine whether this audience would recommend the model, as shown in Figure 16.

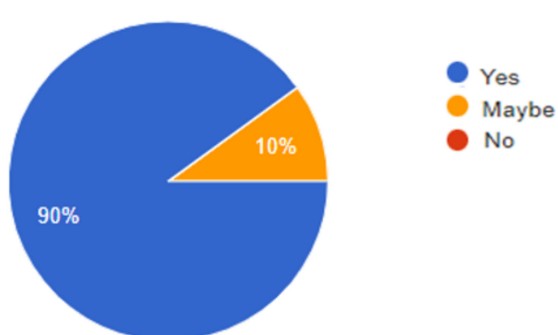

**Figure 16.** Promotion of the model (NPS) for external users.

As a result, only one of the external users indicated they 'might' recommend the gamified model, with the result that 90 percent would, indicating that the model is also in the Zone of Excellence (76–100).

To maintain the attractiveness of the Gamification system in production, an icon in the shape of a 'soccer ball' will be created for the worker's profile, which will be at a visible point on the screen in the event of a long duration. This icon serves the same purpose as the Facebook 'like' button. The worker will have access to five screens and will be able to 'kick the ball' from any of them. When the icon is active, it will change color (from gray and white to black and white).

The 'kick' will remain active until the date changes, which means the worker can only give five 'kicks' per day, enough time for them to consider goals, team performance, and so on. As a result, a single click activates the 'kick', whereas two clicks on the same day deactivate it.

On the Weekly Goals screen, next to each team's avatar, there will be a counter that will increase with each 'kick' of your team. As the number of team kicks increases, this counter will change color, becoming more intense. Every week, the counter will be reset to zero.

A counter will be incremented for each 'kick' taken on the worker feedback screen, next to the week's points badge. As the number of 'kicks' increases, this counter will also change color, becoming more intense. Every week, the counter will be reset to zero.

The audience, who includes people who are not members of the gamification teams, is also welcomed to participate. The audience will be able to react by logging into the system with the 'public profile': next to each team's avatar on the performance screen, they will be able to react, as on Facebook, by selecting the images: 'clapping', 'jumping', and 'fireworks.' Throughout the season, the reactions will be accumulated next to each avatar.

The existence of these icons should not be mentioned to workers in training; the logic should be discovered by them.

These counters will serve as system indicators, measuring both the workers' long-term interest and the workers' participation.

### 4. Final Considerations

There are several issues in the construction industry involving the flow of information between managers and production workers that can be solved or improved by using gamification and visual management on the construction site.

The model received an excellent rating from the worker. According to feedback from construction site trainees, they liked the idea of gamification but stated that they would need system training to better understand it.

The engineers rated the model's quality as excellent and believe it may be viable in construction site production, but it requires a practical application to prove its efficiency. They agreed that the interface is simple and easy to use, and that it has the potential to improve the performance of production teams. As a result, it is possible to conclude that the model is innovative because this practice is not commonly used in construction, and it meets the site's requirements for improving communication with professionals.

External users rated the model's quality as good, easy to understand, and productive, but it must be implemented to demonstrate its efficacy. The interface is interesting, motivating, and playful for them, with well-structured logic. There were some criticisms of the interface presentation, and the system's usability score was average. As a result, for external users, the system can still be improved to achieve excellence. In general, 90% of them would indicate the model to an engineer or builder friend, which proves that they believe in gamification strategies allied to visual management.

The evaluation of this research's participants demonstrates the model's quality, usability, and promotability. Field testing is the only way to prove applicability, viability, and generality.

This qualitative study adds value to the understanding of workers' and engineers' acceptance of using a gamified tool to disseminate production information on the jobsite, but it has the limitation that the behavior of this sample may not reflect the results in similar organizations. As a result, additional research would be required to confirm.

We propose that future work include investigating new data entry interfaces, integrating the BIM (Building Information Modeling) platform into the model, and implementing it on the construction site to see if it works as intended and, based on experience, suggest further adjustments.

**Supplementary Materials:** The following supporting information can be downloaded at: https://www.mdpi.com/article/10.3390/app12115682/s1. The questionnaires used for data collection are available: Worker's questionnaire, Engineer Questionnaire and External Users questionnaire.

**Author Contributions:** Conceptualization, R.M.C.L. and L.R.G.A.; methodology, R.M.C.L.; software, R.M.C.L.; validation, R.M.C.L.; formal analysis, R.M.C.L.; investigation, R.M.C.L.; resources, R.M.C.L.; data curation, R.M.C.L.; writing—original draft preparation, R.M.C.L., L.R.G.A. and I.W.; writing—review and editing, R.M.C.L., L.R.G.A. and I.W.; visualization I.W.; supervision, L.R.G.A. and I.W.; project administration, R.M.C.L. All authors have read and agreed to the published version of the manuscript.

**Funding:** This research received no external funding.

**Institutional Review Board Statement:** The study was conducted in accordance with the Declaration of Helsinki and approved by the Ethics Committee of Centro Universitario SENAI CIMATEC (protocol code no. 4.022.684 dated 12 May 2020) and approved on 13 April 2021 under the Number 4.646.365).

**Informed Consent Statement:** Informed consent was obtained from all subjects involved in the study.

**Data Availability Statement:** Not applicable.

**Acknowledgments:** The authors would like to thank for financial support the National Council for Scientific and Technological Development (CNPq). IW is a CNPq technological development fellow (Proc. 308783/2020-4).

**Conflicts of Interest:** The authors declare no conflict of interest.

## Appendix A

### Worker Questionnaire

What is your name?
How old are you?
What is your occupation?
What is your education level?

1.  Regarding the information available on the screens of the Gamification in Production system, please mark your perception using the UEQ Tool and assign a value from 1 (most negative) to 7 (most positive)

| Negative | 1 | 2 | 3 | 4 | 5 | 6 | 7 | Positive |
|---|---|---|---|---|---|---|---|---|
| obstructive | | | | | | | | supportive |
| complicated | | | | | | | | easy |
| inefficient | | | | | | | | efficient |
| confusing | | | | | | | | clear |
| boring | | | | | | | | exciting |
| not interesting | | | | | | | | interesting |
| conventional | | | | | | | | inventive |
| usual | | | | | | | | leading edge |

https://www.ueq-online.org/

2.  Would you recommend this model to be use into the work you work on?
3.  Do you have any comments or suggestions for improvement on the model?
4.  Do you like to play?
5.  What game do you like to play?
6.  Do you prefer to play in a time or alone? (Alone or in time)
7.  Do you play electronic games (computer, cell phone, video game)?
8.  Do you have and use WhatsApp, Instagram, and Facebook?

**Figure A1.** Worker questionnaire. https://www.ueq-online.org/ (accessed on 13 January 2020).

## Appendix B

### Engineer Questionnaire

What is your name?
How old are you?

1.  Regarding the information available on the screens of the Gamification in Production system, please mark your perception using the UEQ Tool and assign a value from 1 (most negative) to 7 (most positive)

| Negative | 1 | 2 | 3 | 4 | 5 | 6 | 7 | Positive |
|---|---|---|---|---|---|---|---|---|
| obstructive | | | | | | | | supportive |
| complicated | | | | | | | | easy |
| inefficient | | | | | | | | efficient |
| confusing | | | | | | | | clear |
| boring | | | | | | | | exciting |
| not interesting | | | | | | | | interesting |
| conventional | | | | | | | | inventive |
| usual | | | | | | | | leading edge |

https://www.ueq-online.org/

2.  Would you recommend this model to a friend (engineer or technical builder)?
3.  Do you have any comments or suggestions for improvement on the model?

**Figure A2.** Engineer questionnaire. https://www.ueq-online.org/ (accessed on 13 January 2020).

## Appendix C

**External users Questionnaire**

What is your name?
How old are you?
What is your occupation?
What is your education level?

1. Usability/Ease of use of the gamified system (SUS)

| For each statement below, choose an option that represents your perception of the system 0-disagree …. 4-agree | 0 | 1 | 2 | 3 | 4 |
|---|---|---|---|---|---|
| I think I would like to use this system frequently | | | | | |
| I think the system is unnecessarily complex | | | | | |
| I found the system easy to use | | | | | |
| I think I would need the support of a technical person to be able to use this system | | | | | |
| The various functions in this system have been well integrated | | | | | |
| I think there is a lot of inconsistency in this system | | | | | |
| I imagine that most people would learn to use this system very quickly | | | | | |
| I found the system too complicated to use | | | | | |
| I felt very confident using the system | | | | | |
| I needed to learn a lot about context before using this system | | | | | |

**Figure A3.** External users' questionnaire—part 1. Brooke, J. (2013).

2. Regarding the information available on the screens of the Gamification in Production system, please mark your perception using the UEQ Tool.

| Left | 1 | 2 | 3 | 4 | 5 | 6 | 7 | Right |
|---|---|---|---|---|---|---|---|---|
| annoying | | | | | | | | enjoyable |
| not understandable | | | | | | | | understandable |
| creative | | | | | | | | dull |
| easy to learn | | | | | | | | difficult to learn |
| valuable | | | | | | | | inferior |
| boring | | | | | | | | exciting |
| not interesting | | | | | | | | interesting |
| unpredictable | | | | | | | | predictable |
| fast | | | | | | | | slow |
| inventive | | | | | | | | conventional |
| obstructive | | | | | | | | supportive |
| good | | | | | | | | bad |
| complicated | | | | | | | | easy |
| unlikable | | | | | | | | pleasing |
| usual | | | | | | | | leading edge |
| unpleasant | | | | | | | | pleasant |
| secure | | | | | | | | not secure |
| motivating | | | | | | | | demotivating |
| meets expectations | | | | | | | | does not meet expectations |
| inefficient | | | | | | | | efficient |
| clear | | | | | | | | confusing |
| impractical | | | | | | | | practical |
| organized | | | | | | | | cluttered |
| attractive | | | | | | | | unattractive |
| friendly | | | | | | | | unfriendly |
| conservative | | | | | | | | innovative |

https://www.ueq-online.org/

3. Would you recommend this model to a friend (engineer or technical builder)?
4. Do you have any comments or suggestions for improvement on the model?

**Figure A4.** External users questionnaire—part 2. https://www.ueq-online.org/ (accessed on 13 January 2020).

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
