# Peer review of "Visual Management and Gamification: An Innovation for Disseminating Information about Production to Construction Professionals"

_applsci, doi:10.3390/app12115682_

Round 1

Reviewer 1 Report

The paper shows an interesting work, as it proposes a visual tool that is easy and attractive with the aim of improving the performance of construction workers, with studies not higher than high school level.

The tools used to check its application are qualitative tools.

  1. There is a lack of annexes with the questions asked, of which the results have subsequently been obtained. Rather, they only present results about the users: typology, occupation, .....
  2. An introduction to the UEQ-S and NPS tools would be appreciated. What is the difference between these tools? When is it better to use one or the other? Maybe introduce a table with the typology of the questions, evaluation...
  3. In section 3, the numbering of the headings should be revised.
  4. It is important to know what the game consists of and the information that is displayed in order to be able to evaluate it. is it possible to introduce images that give better information about the visualisation of the information that is to be displayed?
  5. What are the limitations of purely qualitative studies?

Author Response

Point 1 - There is a lack of annexes with the questions asked, of which the results have subsequently been obtained. Rather, they only present results about the users: typology, occupation, .....

Response 1 – Thank you for this suggestion, the questionnaires used for evaluation have been added to the Appendix A, Appendix B and Appendix C (Lines 493 - 504).

Point 2 - An introduction to the UEQ-S and NPS tools would be appreciated. What is the difference between these tools? When is it better to use one or the other? Maybe introduce a table with the typology of the questions, evaluation...

Response 2 – The following excerpts have been added to the article to help readers understand the UEQ, UEQ-S, and NPS tools: The UEQ-S tool is a simplified full UEQ model that asks for the evaluation of eight constructs to determine the product's quality according to the user. The product's quality can be measured using a Likert scale ranging from -3 to +3. Products with a quality rating of less than -0.8 are not recommended (Lines 146 -149).

The NSP tool asks the user if they would recommend the product to a friend or colleague. This is an indirect method of determining how much the user trusts the system. This is because no one recommends a product they do not believe into a friend (Lines 159 -162).

Because the research is qualitative, these methods complement one another. The evaluator uses a variety of methods to determine what was good and what could be improved in the system (Lines 177 -179).

Moura Junior, R. Práticas colaborativas gamificadas para Prevenir Lesões por Pressão. Doctoral thesis. Federal University of Bahia, Salvador Bahia Brasil. 2020.

Point 3 - In section 3, the numbering of the headings should be revised.

Response 3 – Thank you very much; the numbering has been revised (Line 182)

Point 4 - It is important to know what the game consists of and the information that is displayed in order to be able to evaluate it. is it possible to introduce images that give better information about the visualization of the information that is to be displayed?

Response 4 – Gamification entails using a soccer metaphor to present the services and goals to be developed and achieved during the week. The system displays the services, quantities, and teams that will develop them on the first screen. The 'Shots on goal' field changes color throughout the week, becoming red for services that are less than 50% complete, yellow for those that are more than 50% complete, and green for those that are more than 90% complete. The second screen shows the teams' performance for four rules, such as safety, production, and so on, as well as a challenge and a mission that will be released at the start of the week.

The team with the most points wins the week's trophy.

The third screen, accessed via a private link, displays the worker's individual performance.

The second and third screens provide feedback to teams and workers on how the services were developed.

Figure 5 depicts how visual management and gamification should be used on the construction site to improve information flow and suggests scenarios in which functions should be used (Lines 290 -308).

Point 5 - What are the limitations of purely qualitative studies?

Response 5 - Thank you for your comment. This qualitative study adds value to the understanding of workers' and engineers' acceptance of using a gamified tool to disseminate production information on the jobsite, but it has the limitation that the behavior of this sample may not reflect the results in similar organizations. As a result, additional research would be required to confirm (Lines 465 -468).

Reviewer 2 Report

In the presented paper, the authors point out the possibilities of using visual management and gamification on construction sites. The aim is to assess the model of a game application, which aims to increase efficiency in the workplace. The results obtained on the basis of questionnaires were presented. I have a few comments on this paper:

- p.7, r.266 - wrong subchapter designation
- p. 12 - Fig. 10 - unify the captions in the picture (everywhere in the pictures is in English)
- p. 12 - the whole subchapter 3.2.3 is missing 

Some questions

1. How long did the individual phases of the introduction of the proposed model last (introduction, training, real use)?
2. The problem with putting game models into common practice is that after some time they lose their attractiveness and may get into a state of stagnation. Individual actors may not give information correctly. How do the authors envisage maintaining attractiveness in the case of a longer-term use of the submitted proposal?

Author Response

Replies to the comments.

- p.7, r.266 - wrong subchapter designation (Line 182).

- p. 12 - Fig. 10 - unify the captions in the picture (everywhere in the pictures is in English) (Line 388).

- p. 12 - the whole subchapter 3.2.3 is missing (Lines 390 -441).

Thank you for your comments. We update the figures and revise and corrected all. 

Point 1 - How long did the individual phases of the introduction of the proposed model last (introduction, training, real use)?

Response 1 – The web system is relatively simple, with only three screens. The trainee was asked to present the system to the workers for evaluation. The presentation lasted approximately 30 minutes. Following that, everyone received the link to navigate the system via WhatsApp. This occurred during the construction site's weekly meeting; at the time, only 10 workers were present due to the pandemic.

The trainee interviewed two workers per day beginning with the next meeting. A form with a consent form, worker profile, technology use, UEQ-S, NSP, and comments and suggestions for improvement was used (Lines 122 -131).

70 engineers and 50 external users (students of computer science, building, engineering, and management, as well as business administrators and engineers) were invited to participate in the evaluation of engineers and external users. Only 15 engineers and 10 users were accepted. The form included the consent form, profile, NSP, as well as comments and suggestions for improvement. Engineers used UEQ-S, while external users used SUS and UEQ complete. We asked participants to respond within a week of receiving the link (Lines 132 -139).

The questionnaires can be found in Appendix A, Appendix B and Appendix C (Lines 493 - 504).

Point 2 – The problem with putting game models into common practice is that after some time they lose their attractiveness and may get into a state of stagnation. Individual actors may not give information correctly. How do the authors envisage maintaining attractiveness in the case of a longer-term use of the submitted proposal?

Response 2 – When it comes to Points, Badges, and Leaderboards (PBL), we agree, but research continues. We interviewed 109 workers for the system's design to determine what theme to use to keep them interested in gamification. Sixty percent of them were familiar with and preferred soccer. This was the motivation behind selecting this story (Lines 187 -189).

According to Schlemmer (2018), PBL is an empiricist educational technique that reduces gamification to a fad, something superficial, and with low innovation power. According to Chou (2019), it is the "shell of a game experience," accounting for only 7% of the total tactics mapped by the Yu-Kai-Chou Octalysis Framework.

According to Alves and Souza (2020), gamification in the PBL perspective impoverishes and limits the possibilities of creating rich, contextualized narratives that reflect content that mobilizes and engages subjects, as in the case of production gamification, where workers enjoy soccer (Lines 90 -97).

We want to put Chou's suggested missions, challenges, and metaphors to the test in order to put this paradigm shift to the test. (Chou, 2019).

Chou, Y. Actionable gamification: Beyond points, badges, and leaderboards. Packt Publishing Ltd, 2019. 145 Pages.

Alves, L.; Souza, M. Westworld: entre no Jogo. In: Sales, M. V. S. Tecnologias digitais, redes e educação: perspectivas contemporâneas. Ed 1. Salvador: EDUFBA, 2020, p. 20-55.

Round 2

Reviewer 1 Report

Thank you for introducing the above suggestions.

Finally, I only have one suggestion left: I have downloaded the new version of the paper and I do not see the annexes indicated in his comments, nor do the comments indicated by the author coincide according to the indicated lines.

Author Response

 Suggestion:

Finally, I only have one suggestion left: I have downloaded the new version of the paper and I do not see the annexes indicated in his comments, nor do the comments indicated by the author coincide according to the indicated lines.

Thank you for this suggestion, the questionnaires used for evaluation have been added to the Appendix A, Appendix B and Appendix C (Lines 527 - 537). As the magazine template does not contemplate attachments, we have placed the questions from the forms as appendices and are sending them to you as supplementary material so that you can get to know them.

Reviewer 2 Report

- r. 267 - subchapter numbering should be 3.2

- p. 13- Table 5, columns "Negative" and "Positive" - values do not match in the column "Negative" is e.g. "easy to learn", then "creative", etc. Need to rework.

I asked 2 questions for the authors and I did not find any answers in the corrected post. Please mark your answers.

I repeat

1. How long did the individual phases of the introduction of the proposed model last (training, real use)

2. The problem with putting game models into common practice is that after some time they lose their attractiveness and may get into a state of stagnation. Individual actors may not give information correctly. How do the authors envisage maintaining attractiveness in the case of a longer-term use of the submitted proposal?

Author Response

Replies to the comments.

- r. 267 - subchapter numbering should be 3.2

- p. 13- Table 5, columns "Negative" and "Positive" - values do not match in the column "Negative" is e.g. "easy to learn", then "creative", etc. Need to rework.

Thank you for your comments. The snippet has been moved to subchapter 3.2, (Lines 285-313) and the headings in table 5 ('Left' and 'Right' for the full WEQ) have been corrected (Line 411). We update the Table 5 and revise and corrected all. 

Note: There has been a change in the final considerations - (Lines 496-498) ‘The evaluation of this research's participants demonstrates the model's quality, usability, and promotability. Field testing is the only way to prove applicability, viability, and generality.’

 Point 1 - How long did the individual phases of the introduction of the proposed model last (introduction, training, real use)?

Response 1 – The trainee was assigned to conduct the training and interviews. In this same company, he was a member of the research team during the data collection phase.

The model was presented to the workers, demonstrating the tool's functionality, the meaning of the icons, team feedback, and individual feedback. The training lasted approximately 30 minutes. Following that, everyone was given a WhatsApp link to use the system. This occurred following the weekly construction site meeting, during which the trainee was available to answer any questions about the system. Beginning with the next weekly meeting, the trainee interviewed two workers per day, gathering their feedback on the model using a form that combined the UEQ-S and NPS tools (Figure A). The interviewer read the form's questions and alternative answers and marked the one selected by the interviewee. Only ten workers were performing regular services at this time due to the pandemic and the mandatory reduction in the number of employees on site to keep the distance.

70 engineers and 50 external users (computer science, building, engineering, and management students, business administrators and engineers) were invited to participate in the engineers and external users’ evaluation. Only 15 engineers and ten outside users agreed. A link to the system was made available for this public in the system's public characteristics as well as in the private links of a fictitious functionary, for model analysis and information flow. Because the system is simple and clear, the training was limited to a brief explanation of tool's functionality and the meaning of the icons.

Engineers were also instructed to complete the questionnaire using the UEQ-S and NPS tools following system testing (Figure B), (Lines 122 -143).

All participants were given one week to use the system and complete the questionnaires (Lines 150 -151).

The questionnaires can be found in Appendix A, Appendix B and Appendix C (Lines 527 - 538).

Point 2 – The problem with putting game models into common practice is that after some time they lose their attractiveness and may get into a state of stagnation. Individual actors may not give information correctly. How do the authors envisage maintaining attractiveness in the case of a longer-term use of the submitted proposal?

 Response 2 - To maintain the attractiveness of the Gamification system in production, an icon in the shape of a 'soccer ball' will be created for the worker's profile, which will be at a visible point on the screen in the event of a long duration. This icon serves the same purpose as the Facebook 'like' button. The worker will have access to five screens and will be able to 'kick the ball' from any of them. When the icon is active, it will change color (from gray and white to black and white).

The "kick" will remain active until the date changes, which means the worker can only give five "kicks" per day. Enough time for him to consider goals, team performance, and so on. As a result, a single click activates the 'kick,' whereas two clicks on the same day deactivate it.

On the Weekly Goals screen, next to each team's avatar, there will be a counter that will increase with each 'kick' of your team. As the number of team kicks increases, this counter will change color, becoming more intense. Every week, the counter will be reset to zero.

A counter will be incremented for each 'kick' taken on the worker feedback screen, next to the week's points badge. As the number of ‘kick’s’ increases, this counter will also change color, becoming more intense. Every week, the counter will be reset to zero.

The audience, who includes people who are not members of the gamification teams, is also welcomed to participate. The audience will be able to react by logging into the system with the 'public profile': next to each team's avatar on the performance screen, they will be able to react, just like on Facebook, by selecting the images: 'clapping,' 'jumping,' and 'fireworks.' Throughout the season, the reactions will be accumulated next to each avatar.

The existence of these icons should not be mentioned to workers in training; the logic should be discovered by them.

These counters will serve as system indicators, measuring both the workers' long-term interest and the workers' participation (Lines 448 - 474).